# Water as a Structural Marker in Gelatin Hydrogels with Different Cross-Linking Nature

**DOI:** 10.3390/ijms252111738

**Published:** 2024-10-31

**Authors:** Yuriy F. Zuev, Svetlana R. Derkach, Ivan V. Lunev, Alena A. Nikiforova, Mariya A. Klimovitskaya, Liliya R. Bogdanova, Polina V. Skvortsova, Rauf Kh. Kurbanov, Mariia A. Kazantseva, Olga S. Zueva

**Affiliations:** 1Kazan Institute of Biochemistry and Biophysics, FRC Kazan Scientific Center, Russian Academy of Sciences, Lobachevsky Str. 2/31, 420111 Kazan, Russia; lounev75@mail.ru (I.V.L.); alnikiforova22@gmail.com (A.A.N.); mklimovitskaya@mail.ru (M.A.K.); skvpolina@gmail.com (P.V.S.); kurbanov@kibb.knc.ru (R.K.K.); masha353kazan@gmail.com (M.A.K.); 2Laboratory of Chemistry and Technology of Marine Bioresources, Institute of Natural Science and Technology, Murmansk Arctic University, 183010 Murmansk, Russia; derkachsr@mstu.edu.ru; 3Kazan Federal University, Kremlevskaya Str. 18, 420008 Kazan, Russia; 4School of Applied Mathematics, HSE University, Tallinskaya Str. 34, 123458 Moscow, Russia; 5Institute of Electric Power Engineering and Electronics, Kazan State Power Engineering University, Krasnoselskaya Str. 51, 420066 Kazan, Russia; ostefzueva@mail.ru

**Keywords:** gelatin hydrogels, physical and chemical crosslinking, state of water, gel structure

## Abstract

We have studied the molecular properties of water in physically and chemically cross-linked gelatin hydrogels by FTIR-spectroscopy, NMR relaxation, and diffusivity and broadband dielectric spectroscopy, which are sensitive to dynamical properties of water, being a structural marker of polymer network. All experiments demonstrated definite reinforcement of the hydrogel net structure and an increase in the amount of hydrate water. FTIR experiments have shown that the chemical cross-linking of gelatin molecules initiates an increase in the collagen-like triple helices “strength”, as a result of infused restriction on protein molecular mobility. The “strengthening” of protein chains hinders the mobility of protein fragments, introducing complex modifications into the structural properties of water which are remained practically unchanged up to up to 30–40 °C.

## 1. Introduction

Gelatin is the outstanding alimentary protein obtained by the hydrolytic splitting of fibrillar collagen proteins, which is the constituent of bones, cartilages, connective tissue, and skin of mammals and fishes [1,2,3]. Gelatin occupies an important place among the natural hydrocolloids in modern nutritional, pharmaceutical, and some other technologies [4,5,6,7].

Undoubtedly, the basic technological property of gelatin is its gelling, i.e., the ability to form thermo-reversible hydrogels [8]. This is the consequence of gelatin’s primary and secondary structures that are built from collagen-like polypeptide α-chains and triple helices [9]. Approximately one-third of the amino acids in gelatin α-chains contain glycine (Gly) as a part of the repeating tripeptide Gly–X–Y, where X, most often, is proline (Pro) and Y is hydroxyproline (Hyp). The Gly–X–Y triads in the gelatin polypeptide chains are the main structural players in the formation of collagen-like triple helices [10]. Gelatin, when jelling at temperatures below 30 °C, follows a number of stages [1,11]. The first stage is the transition of the macromolecular conformation from a coil to a collagen-like helix. During the second phase, helices aggregate with disordered chains, forming a spatial protein network that is called hydrogel due to the absorption of a significant amount of water in its volume.

Polymer/biopolymer hydrogels can be of two main types, “physical” hydrogels with non-covalent cross-linking and “chemical” ones with intra- and inter-polymer covalent bonding. Traditionally more prevalent are physically cross-linked hydrogels, in which the spatial polymer network exists due to the mechanical entanglement of macromolecules and their intra- and inter-molecular interactions, such as ionic bridges, hydrogen bonding, and hydrophobic forces [12,13,14]. Gelatin “physical” hydrogels, exhibiting dual liquid and solid properties, play a crucial role in the food and pharmaceutical industries due to their biocompatibility and unique technological potential [15,16].

Recently, there has been a steep rise in novel interest in gelatin hydrogels in the form of so-called “jelly ice cubes” [17]. Usually, to create a cooling effect and maintain low temperatures for food storage, ice is widely used. However, under repeated freeze–thaw cycles, the melt water becomes a convenient medium for the spreading of bacteria. Therefore, the demand for safe and reusable cooling systems as ice substitutes is very high. The alternative cooling systems must have a high heat absorption capacity in comparison with traditional ice, and have sufficiently high mechanical strength to ensure repeated use [18]. In this regard, hydrogels containing up to 90% water are very promising systems, in particular naturally available and biocompatible gelatin-based ones.

The usual gelatin hydrogels are stabilized mainly by intermolecular and intramolecular hydrogen bonds formed during the process of sol−gel transition. However, during deep-freeze, these H-bonds can be easily disrupted by phase separation during the formation of large-sized ice grains. Not long ago, gelatin was proposed for the engineering of novel cooling systems, partially because it carries a high antimicrobial activity [17]. This novel technology involves a rapid-freezing, slow-thawing treatment and a subsequent photo-cross-linking reaction induced by a newly discovered photosensitizer—menadione sodium bisulfite (MSB) [17,18,19]. The developed freezing–thawing treatment results in the formation of a more uniform gelatin net before the gelatin undergoes chemical bonding. The use of physical cross-linking in combination with MSB chemical bonding imparts stable mechanical properties to hydrogels, which ensures their structural stability during temperature fluctuations and phase transitions of water. In addition, MSB brings antimicrobial defense to gelatin hydrogel [18]. Unfortunately, at this moment there is not enough information about the molecular origin of gelatin hydrogel obtained with the proposed procedure.

One can suppose the corresponding alteration of gelatin and water properties under the changes properties under formation of gelatin hydrogel structure. It is obvious that the physico-chemical characteristics of water can provide unique information about the structural state of gelatin-based hydrogels made for use as a new type of food cooling medium. Thus, the main goal of this study was to study the main structural features of a new type of food coolant, “jelly ice cubes” [17,18,19], i.e., chemically cross-linked gelatin-based hydrogels, in comparison with initial “physical” gelatin hydrogels. We have studied the molecular properties of water in “physically” and “chemically” cross-linked gelatin hydrogels by FTIR-spectroscopy, NMR relaxation and diffusivity, and broadband dielectric spectroscopy, which are sensitive to dynamical properties of water being a signature of polymer net structure.

## 2. Results and Discussion

### 2.1. Current Conceptions on Geletin Gels Structure

The routine model of the physical network in gelatin gel is a combination of disordered polypeptide fragments with ingrained triple intermolecular collagen-like helices, which play the role of junction zones in the gel network (Figure 1a). Different types of non-covalent bonding are involved in the formation and stabilization of the 3D physical network of gelatin gel: inter- and intra-molecular hydrogen bonds, and electrostatic and hydrophobic interactions between branched-chain amino acids [11,20]. Triple helices are distributed among disordered fragments of the polypeptide chain to achieve the lowest possible free energy [21]. The physical interactions between macromolecules remain relatively stable and strong in the temperature range between the melting point of ice and the gel melting point. Gelatin hydrogels contain large amounts of water that is strongly confined in the protein twisted network.

Under decreases in temperature up to the water freezing point, the resulting large ice crystals cause serious damage to junction zones of the physical network. When hydrogel is defrosted, the swelling results in the appearance of disordered domains inside and between macromolecules. Recently, a novel procedure was proposed [18,19] which alters these undesirable phenomena and increases the mechanical strength and cold-capacity of gelatin hydrogel by reinforcing of its network structure. The sequential treatment of hydrogel with a rapid freezing–slow thawing process, accompanied by photo-induced cross-linking (chemical covalent cross-linking) demonstrates a synergistic effect on structural strengthening [19].

The formation of a chemically cross-linked gelatin gel network can be represented as follows. The rapid freezing of water in hydrogel (in the form of small ice grains) leads to an increase in the local concentration of “dissolved” gelatin and the formation of more ordered, densely packed local protein structures, which are physically cross-linked microcrystalline domains [18]. Thus, the number of cross-linked junctions between the gelatin molecules in such zones is increased compared to that in conventional hydrogel. Microcrystalline domains are in contact with amorphous regions of the gelatin chains, in which macromolecules are also less flexible due to their dense packing. It is important that, after the stage of slow thawing and the transition of water into the liquid state, the dense packing of gelatin chains in crystalline domains and the overall degree of the chains’ freedom remain.

To stabilize the locally formed, close-packed 3D structure with restricted molecular motion, to increase its mechanical strength, and to inhibit the decrease in physical cross-linking caused by the swelling of gelatin chains during thawing, the chemical cross-linking is carried out using a photo-induced chemical reaction with menadione sodium bisulfite (MSB). The photo-cross-linking reaction results in the formation of covalent bonds between gelatin molecules at the intersections of pseudo-crystalline and amorphous regions within the gelatin matrix [19], maintaining and fixing the pre-formed modified gelatin network (Figure 1b). The chemical interaction between gelatin macromolecules occurs via a radical mechanism. When irradiated with an appropriate UV light source, the photosensitizer MSB is excited to the triplet state [22], abstracting hydrogen atoms (mainly hydrogens attached to α-carbons of gelatin amino acids and specific side groups in polypeptide chains) from gelatin molecules, generating MSB and protein radicals for which chemical binding leads to the covalent linking of gelatins [19]. As a result, one can obtain reinforced gelatin hydrogel which opposed to the action of increased temperatures. (Figure 2).

As a consequence of rapid freezing–slow thawing, accompanied by photo-induced covalent cross-linking, the newly organized structure of gelatin hydrogel is formed with local close pseudo-crystalline domains that are uniformly distributed along the hydrogel volume. Upon the chemical cross-linking, a local increase in the density of gelatin mesh elements (binding nodes) occurs, with an increase in the spacing between these dense elements. One can see our SEM data in the following figure (Figure 3), where the increase in “pore volume” between elements for the “chemical” hydrogel is detected in comparison with the “physical” hydrogel. We supposed that such morphological alterations are the result of changes in the gelatin and water supramolecular structure and applied a number of physical-chemical methods to analyze some details of this phenomenon.

### 2.2. Protein and Water Molecular Properties in “Physical” and “Chemical” Hydrogels

To start our data analysis, let us look at the FTIR data. FTIR spectroscopy is rather successfully employed to analyze the state of water in protein systems [23,24] and to study the structural features of different proteins [25,26], including gelatin [27]. The spectral pattern within the Amide I band describes the conformational structure of gelatin polypeptide chains [28]. It is known that Amide I is coupled with the C=O stretching vibration, connected to the protein CN-stretch, and Amide II is sensitive to the hydrogen bonding of N-H groups in the range of 1700–1500 cm^−1^ [29].

In Figure 4a,b, one can see that the character of the temperature spectral changes is qualitatively identical for the original and cross-linked hydrogels. This means that, in general, the structural states of the systems are close and can respond similarly to increases in temperature.

However, one can detect some structural alterations, which determine the temperature stability of the studied systems. The temperature destruction of collagen-like triple helices is accompanied by the cleavage of hydrogen bonding N-H(Gly)...O=C(Pro), which is reflected in the shift of the Amide II band maximum (N-H bending vibrations) shown in Figure 4c. One can see also the results for the structural degradation of collagen from rat tail (Corning collagen Type 1 from rat tail, CLS354236, 3.62 mg/mL in 0.02 M acetic acid), determined by us early. The denaturation temperature is determined by the inflection of the dependence under saturation. It is obvious that the limit positions of the Amide II peak for the studied gelatins at 50 °C are far from the level of collagen denaturation, thus meaning that the denaturation of gelatins at this temperature is incomplete. However, one can note the higher thermal stability of the chemically cross-linked gelatin compared to the original one (moderate shift in adsorption to higher temperatures), despite the fact that the level of helicity at 4 °C of the first one is lower (frequency is lesser). In other words, the presence of intra- and inter-molecular chemical shifting results in a moderate increase in the collagen-like triple helices’ “strength”. Possibly, the existence of cross-linking chemical bonds nearby triple helices hinders, to some extent, their “untwisting”.

Another comparative characteristic of studied hydrogels is the state of confined water in hydrogels. To study the properties of the hydrogel water in the absorption band of stretching OH vibrations, we analyzed the differential FTIR spectra (3800–3000 cm^−1^), subtracting the spectrum of pure water from the spectrum of the original gelatin gel and the spectrum of a 1% MSB water solution from the spectrum of the cross-linked gelatin hydrogel at current temperatures. This approach permitted us to view the alteration of water properties in the gelatin hydrogel in comparison with bulk water. Under an increasing temperature, in the region of 3400–3200 cm^−1^, the negative difference between the water absorption in the bulk water and in the hydrogels increased (Figure 5a). This band is responsible for the OH stretching vibrations of water with three and four hydrogen bonds, characteristic of the unperturbed structure of bulk liquid water. Thus, from these data, one can see the alterations in the structure of water confined in hydrogels in comparison with the bulk water. From Figure 5a, it follows that, for the original hydrogel at 4 °C, the difference in water structure is minimal, increasing with temperature growth. Therefore, the heating destroys the structure of water in the presence of a gel. Unlike the original gel, in the chemically cross-linked one, a significant difference in water state is observed already at 4 °C, but upon subsequent heating the changes are less pronounced. In other words, chemical cross-linking results in initial modifications of water structural properties, and the structure of cross-linked hydrogel retains these changes up to 30–40 °C. The stretching vibrations of water OH groups are intra-molecular and characterize the state of individual water molecules. The absorption spectra are the statistical sum of the deposits from the bulk and hydrated water. Difference spectra characterize the interactions between protein hydration centers and individual water molecules. The intensity of the difference spectrum is proportional to the number of these populations. The reason why the original and cross-linked gels have different effects on the water structure may be due to the different mobility of the gelatin peptide chains. The heating of the original gel causes the unfolding of helices and the expansion of individual chains throughout the entire available volume, causing the maximum accessibility of protein polar groups for the water. In the cross-linked gel, the swelling is significantly limited and the accessibility of protein polar groups for water changes only slightly.

Unlike stretching vibrations, the librational motions (rotational swings around equilibrium position [30]) characterize collective mobility of water molecules in the bulk [31]. The bater band available for study at 2130 cm^−1^ is the combination of the deformations and librational vibrations. The OH stretching vibrations are intramolecular in nature, and the temperature has a small effect on the energy of these vibrations, while the nature of the librational movement significantly depends on the exchange of energy between molecules, and the contribution of these vibrations to the temperature dependence of the band is decisive [32,33,34]. The long-wavelength shift of the position of the maximum of this band (Figure 5b) reflects a decrease in the elasticity of the hydrogen bond network due to its thermal expansion. From this point of view, gelatin gel weakens collective interactions between water molecules, loosening the network of hydrogen bonds. This effect in gelatin gels at 4 °C is relatively small and, within the error limits, does not depend on the presence of cross-linking. However, with temperature increases, the difference in the dynamics of water molecules in chemically cross-linked gel differs quite sharply from those of pure liquid and original gel. It can be assumed that, in the original gel, the macromolecular chains are sufficiently mobile, without significant obstacles to the collective movement of water molecules and to their mutual adjustment during heating. In contrast, the presence of cross-links limits the mobility of protein fragments, which can introduce disturbances into the dynamics of water, which is especially evident during heating.

Thus, our FTIR experiments have shown that the chemical cross-linking of gelatin gel leads to an increase in the collagen-like triple helices’ “strength” as a result of the infused restriction of protein molecular mobility. The “strengthening” of protein chains hinders the mobility of protein fragments, thus introducing complex modifications of water structural properties and retaining them from the disturbing action of temperature up to 30–40 °C.

For the further analysis of the molecular properties of “physical” and “chemical” hydrogels, we applied relaxation and self-diffusion NMR experiments. In the present study, we used the spin–lattice (T_1_) and spin–spin (T_2_) relaxation and translation self-diffusion coefficients (D) of water, the dynamical properties of which are the structural and interaction signatures of complex molecular systems [35,36,37]. The water–protein interactions display the details of protein structural states [38,39,40,41,42], e.g., the after-effects of protein gelation [43]. The water molecular mobility in biopolymer networks is determined by the presence of a considerable number of macromolecular interfaces, providing a hydrophobic effect and hydrogen bonding for water, which constrains the water molecular motion [44,45,46].

Figure 6 shows the temperature dependences of T_1_, T_2_, and D for both studied systems. The original and chemically cross-linked hydrogels are characterized by single T_1_ and T_2_ relaxation times for their water protons. In spite of the existence of free and bound water fractions, typical for different protein–water systems [43,47,48], in such systems, usually only one spin–lattice and one spin–spin relaxation time are observed due to a rapid exchange between water fractions in protein-based systems [49,50]. For water protons with sufficient mobility, one can consider the following correlation [37,43]:T_1_, T_2_~1/τ_c_,(1)
where τ_c_ is the correlation times for the random motion of water. T_1_ and T_2_ reflect the mobility of water molecules in such a way that the restriction of the water mobility decreases the value of the relaxation time [51]. In other words, the decrease in T_i_ signifies the slowing down of the water mobility. Besides contributions from free and bound water, there is also another one from magnetization transfer between the water and accessible protein protons through chemical exchange and/or cross-relaxation [50]. In addition, in chemically cross-linked hydrogel, we are faced with one more factor, which influences water mobility and relaxation rates. This factor appears to be due to the presence of a sufficient amount of sodium ions in the cross-linked system, since we have used a menadione sodium bisulfite (MSB) as the coupling agent. It is the sodium salt and the source of a sufficient amount (1%) of free Na^+^ ions in the hydrogel bulk, which is known as an active kosmotrope towards water [52]. Na^+^ ions form a tight hydration shell of from three to five water molecules, held electrostatically around the alkali metal ion [53,54]. Thus, a sufficient amount of water molecules is bound to sodium ions, decreasing their average mobility and the total relaxation rate.

All the abovementioned water fractions and relaxation mechanisms give certain contributions to the relaxation rate (time) through the existence of multiple correlation times for each of the exchanging water species. While the analysis of the T_1_ relaxation gives information about more fast movements, T_2_ provides insight into the long correlation time (slow movements). According to the previously obtained information on water properties in “physical” biopolymer hydrogels [37], T_1_ displays the water relaxation rate, which, in the framework of two-site models [55,56], can be expressed as the weighted average sum of contributions from the mobility of free and bound water:1/T_1_ = p/T_1W-B_ + (1 − p)/T_1W-FREE_,(2)
where p is the relative portion of water in the bound state, and T_1W-B_ and T_1W-FREE_ are the spin–lattice relaxation times of bound and free water. The two-site model is specifically used in NMR experiments, having a rather long observation time in comparison with the life time of the detected molecules in different microenvironments, in comparison with exchange rate between them [55,56]. The decrease in T_1_ (increase in τ_c_ according to correlation 1), depicted in Figure 6a, is evidence of the deceleration of the apparent water mobility in the chemically cross-linked gel, being the consequence of an increase in the bound water content (Equation (2)). We have analyzed the obtained dependencies using Equation (2), assuming as a first approximation that T_1W-B_ = 0.1 T_1W-FREE_ [57] and using the T_1W-FREE_ values for pure water [58]. It turned out that the amount of bound water in a chemically cross-linked gel can increase by almost 15 times compared to the original sample, from 0.33% to 4.8% of the total amount of water.

The T_2_ NMR relaxation behavior, which is also sensitive to slow movement, also depicts the effects of magnetization transfer between water and gelatin protons, which may occur through chemical exchange and/or cross-relaxation [37]. Therefore, the weighted average sum of the main contributions from the exchangeable water fractions to the spin–spin relaxation time, T_2_, can be presented as
1/T_2_ = p_1_/T_2W-B_ + p_2_/T_2W-FREE_ + p_3_/T_2-CROSS_,(3)
where T_2W-B_ and T_2W-FREE_ are the spin–spin relaxation times of the bound and free water, T_2-CROSS_ is the cross-relaxation term, and p_i_ is the relative portion of every relaxing population (p_1_ + p_2_ + p_3_ = 1). The general finding from the T_2_ results is also an overall decrease in water mobility in the chemically cross-linked gelatin hydrogel in comparison with the original one (Figure 5b), which is visibly greater than the effect of sodium hydration, shown by a green asterisk. The water exists in a bulk and in a motion-confined state due to interaction with protein, with very fast exchange between them. In addition, the water, bound to protein, undergoes proton chemical exchange with the protein. Therefore, the T_2_ is a weighted average of these fractions, resulting in the experimental T_2_. During heating, the protein alters the number of polar groups exposed to water [59,60], which was also shown by the FTIR results, and modulates the chemical exchange rate with adjacent water molecules, determining the temperature dependence [43]. It should be noted that our experiments demonstrate identical temperature profiles for the original and cross-linked gelatin gels.

Figure 6c presents one more NMR result for the water mobility in the original “physical” gelatin hydrogel and its alternative, “chemical” or covalently cross-linked one. In contrast to the T_1_ and T_2_ relaxation, having a rather complicated interrelation with water molecular motion, complicated in addition by magnetization phenomena between the water and gelatin protons, the self-diffusion coefficient obtained in the NMR experiment directly characterizes the molecular diffusivity of water. Although, due to the specifics of the NMR experimental timescales, one also obtains the weighted average sum of the contributions from water diffusion mobility in the bound (D_BOUND_) and free (D_FREE_) populations, with fast exchange between them [61,62]:D_W_ = pD_BOUND_ + (1 − p)D_FREE_,(4)
where p is the related portion of bound water. For simplicity, we combined all bulk water into one family and the bound water into the second one. In Figure 6c, one can see the parallel slowing of the apparent water diffusive mobility in the chemically cross-linked hydrogel relative to the original one. Most likely, this is the consequence of (1) an increase in the bound water relative portion due to some protein unfolding and increases in the number of polar groups exposed to water, as follows from the FTIR data; (2) an increase in obstructions to the water translation motion [63] due to additional strengthening of the gelatin net; and (3) the hydration of free Na^+^ counter ions as a result of dissociation of the coupling agent menadione sodium bisulfite. The activation energy, 16.2 ± 0.5 kJ·Mol^−1^, of the water translational diffusion, obtained from self-diffusion data, is very close for both systems. This value correlates well with the activation energy for the self-diffusion of pure water, E_W_ = 17.8 kJ·Mol^−1^ [64], signifying that most of the water in hydrogels has similar properties to pure bulk water. The closeness in the order of these values verifies our assumptions about fast exchange between water populations and a low volume of bound water in comparison with the total water content in hydrogels.

Accordingly, the complex of complementary NMR data support, in general, the results of our FTIR study. We observed the slowing of water mobility in the chemically cross-linked system, which, in our opinion, appears due to the meshing of the hydrogel net and light disturbance of the protein structure.

Another popular physical-chemical approach to study the properties of water in different water–biopolymer systems is broad-band dielectric spectroscopy [38,40,65]. Here, the water relaxation properties act strikingly, reflected in the contrast signature of the structural and dynamic properties of the bounding matrix. We applied this method to estimate the difference between the original and cross-linked gelatin hydrogels. Figure 7 depicts the water response to the structural differences in the studied systems. Figure 6a shows the result of the least-squares procedure, applied to the regression of the experimental data, showing that complex dielectric spectra of gelatin hydrogels can be represented satisfactorily by the sum of two relaxation processes (Figure 7a):(5)$$\varepsilon^{*}=\varepsilon_{\infty }+\frac{\Delta \varepsilon_{1}}{1+{\left(i\omega \tau_{1}\right)}^{\alpha_{1}}}+\frac{\Delta \varepsilon_{2}}{1+{\left(i\omega \tau_{2}\right)}^{\alpha_{2}}},$$
where $\Delta \varepsilon_{i}$ is the increment of the dielectric constant, $\tau_{i}$ is the relaxation time, $\alpha_{i}$ is the distribution of the relaxation times (0 < α < 1), $\varepsilon_{\infty }$ is the high-frequency dielectric constant, and ω is the circular frequency. These processes correspond to the relaxation of water (about 10 GHz) and gelatin (10–100 MHz) [66,67]. Our results show a low-frequency shift of both processes in the presence of chemical cross-linking, complying with the overall slowing down of molecular motion. At present, there are no clear insights into the mechanisms of low-frequency relaxation, which could be the consequence of different polarization processes of gelatin molecules and network interfaces [66,68]. Therefore, let us discuss in more detail the water process only.

Figure 7b depicts the decrease in the amount of free bulk water in the hydrogels. The dielectric strength or amplitude of the relaxation process, 8ε, is the measure of the dielectric polarizability, proportional to the number of identical dipoles. In our case (GHz frequency range), they are the free water molecules, and lowering of the amplitude shows a decrease in free water. One can see that “chemical” hydrogel is characterized by a higher level of water hydration. The general deceleration of water mobility in the chemically cross-linked system is demonstrated also by the increase in the relaxation time (Figure 7c). Further, one can see a definite increase in the activation energy of the water relaxation process from 16 kJ·Mol^−1^ for pure water (15.9 kJ·Mol^−1^ from [69]) to 17.5 kJ·Mol^−1^ for “physical” and 18.1 kJ·Mol^−1^ for “chemical” gelatin hydrogels, confirming the results from the NMR experiments.

## 3. Materials and Methods

### 3.1. Chemicals and Sample Preparations

Aqueous solutions of 12.5% porcine gelatin (Type A, 300 bloom food grade, Sigma, G2500, Sigma-Aldrich, St. Louis, MO, USA) and 5% menadione sodium bisulfite (MSB) (Sigma, M5750) were prepared separately using deionized Milli-Q water purified with the “Arium mini” ultrapure water system (Sartorius, Gottingen, Germany). Initially, samples of porcine gelatin were swelled at room temperature overnight, then they were kept for 1 h at 50 °C under ultrasonic treatment in Bandelin SONOREX TK52 ultrasonic bath (Bandelin, Berlin, Germany, 100 W, 35 kHz) until fully dissolved. After that, MSB solution was added to one of the gelatin solutions. The final solution, where the concentrations of MSB and gelatin were 1% and 10%, respectively, was mixed at 40 °C for 1 h and further used to prepare “chemically” cross-linked gelatin hydrogel, according to known formulations [17,19].

To prepare original hydrogel aqueous, solutions of 10% porcine gelatin were swelled at room temperature overnight then kept for 1 h at 50 °C under ultrasonic treatment until fully dissolved.

The obtained gelatin hydrogel (original) and gelatin–MSB system (chemically cross-linked) were settled in silicon molds (mostly in 10 × 10 × 10 mm mold unless specified) to complete the sol−gel transition under 4 °C overnight. Then, it was treated with one freeze−thaw cycle that consisted 18 h of freezing at −36 °C and =6 h of thawing at 4 °C. Once ejected from the mold, the sample was irradiated in a UVA cross-linking chamber (Rayonet RMR-600, Southern New England) equipped with five UVA lamps (350 nm, 4 W) on the ice surface to maintain the chamber temperature below 30 °C. Samples were stored at 4 °C for preservation.

### 3.2. Scanning Electron Microscopy

The morphology of freeze-dried samples of gelatin “physical” and “chemical” gels was analyzed by means of scanning electron microscopy (SEM) with the help of the field emission scanning electron microscope “Merlin” (“Carl Zeiss”, Oberkochen, Germany). Surface morphology was studied at accelerating voltage of 5 kV. The experiments were carried out using gelatin cryogels [70,71] prepared as follows. The prepared gelatin gels were left overnight at 4–6 °C. Then, samples were frozen in liquid nitrogen and vacuum freeze-dried to obtain xerogels. The fractured xerogel sections were bespread with gold/palladium (80/20) for SEM observations. The SEM experiments were carried out in the Interdisciplinary Center for “Analytical Microscopy” (Kazan Federal University, Kazan).

### 3.3. FTIR Spectroscopy

The Fourier transform infrared (ATR-FTIR) experiments were fulfilled to study the gelatin and water structure in gels. The spectra were registered using a FTIR spectrometer Invenio S (Bruker, Preston, UK) equipped with the attenuated total reflection (ATR) accessory with a triple-bounced ZnSe crystal. The spectra were collected at a 4 cm^−1^ resolution, accumulating 128 scans in the range of 4000–800 cm^−1^. Besides the “physical” and “chemical” gelatin hydrogels, we prepared an additional “physical” sample, containing 1% MSB (without UV exposure to radiation), to take into account spectral deposit of the coupling agent. Gel samples, stored at 4 °C, were placed on the ATR crystal, preliminarily refrigerated to 4 °C. The measurements were made in series at 4, 10, 20, 30, 40 and 50 °C. The resulting spectra were corrected for solvent water and atmospheric water vapor absorption. The spectra were processed and the second derivative was calculated using the OPUS 7.0 program (Bruker Optik GmbH 2012).

### 3.4. NMR. Relaxation and Self-Diffusion Masurements

The water ^1^H NMR relaxation and self-diffusion experiments were carried out using Bruker AVANCE III WB 400 NMR spectrometer (Bruker, Preston, UK) with a working frequency of 400.27 MHz for ^1^H. Self-diffusion experiments were conducted with the help of Diff50 Pulsed Field-Gradient (PFG) probe, using the stimulated echo pulse sequences DiffDste. The length of the 90° pulse was 8.12 µs, δ was in the range of 1.5 ms, and the amplitude of magnetic gradient g was varied from 0.04 up to 20 T·m^−1^. The recycle delay was 10 s. All measurements were carried out during cooling in the temperature range from 50 to 5 °C. The equipoise time for thermal equilibrium of sample volume was 10 min. The water T_1_ and T_2_ relaxation time measurements were performed with the inversion-recovery (180°-τ-90°-fid) and CPMG (90°-τ-180°-τ-echo) pulse sequences, respectively. The value of τ was 2 ms, with 128 points on relaxation decay. Data were processed with the help of the Bruker Topspin 3.5 software (Bruker Biospin Corporation, Billerica, MA, USA).

Water self-diffusion coefficient and T_2_ relaxation in 1% NaCl water solution were obtained using Bruker AVANCE III 600 MHz spectrometer equipped with triple resonance TBI probe, and z-gradient. Self-diffusion coefficient was measured using the standard “stimulated echo” pulse sequence with bipolar gradient pulses stebpgp1s19 (Δ = 50 ms, δ = 16 ms). The Topspin software was used for data processing and analysis. For chemical shift calibration, the glass coaxial insert in NMR sample tubes containing solution of TSP ((3-(trimethylsilyl)-2,20,3,30-tetradeuteropropionic acid) sodium salt) in D_2_O was used.

### 3.5. Dielectric Spectroscopy

The dielectric experiments, in about 8 frequency decades, were carried out in three stages (Figure 8). In the first stage, dielectric measurements were fulfilled in the frequency range of 100 MHz–60 GHz from 50 °C to 0 °C (cooling). Dielectric spectra were obtained with the PNA-X Agilent N5247A network analyzer (Agilent Technologies, Santa Clara, CA, USA) in accordance with protocol of the built-in software package Agilent 85070. The temperature was maintained using a LOIP LT 900 thermostabilizer with a step of 5 °C and temperature maintenance accuracy ±0.1 °C. A coaxial performance probe with a diameter of 10 mm was used as the measuring cell. The sample exposure time at each temperature was 10 min.

At the second stage, the measurements were carried out in the frequency range of 1 MHz–1 GHz on the Agilent E4991A Frequency Response Analyzer as a part of the Novocontrol BDS-80 measuring complex. The sample temperature was maintained using the Quatro system, in a temperature range from 50 °C to 0 °C with steps of 5 °C and an accuracy of ±0.5 °C.

At the third stage, the measurements were fulfilled in the frequency range of 1 Hz–10 MHz using the Alpha Frequency Response Analyzer, being a part of the Novocontrol BDS-80 measuring complex. Thermal stabilization was carried out using the Quatro system, in the temperature range from 50 °C to 0 °C, in steps of 5 °C with an accuracy ±0.5 °C.

The frequency ranges of the three measurements overlap, which made it possible to combine the measured spectra and obtain a broadband spectrum in the frequency range of 1 Hz to 60 GHz for each sample and measured temperature (Figure 7). The dielectric parameters of the spectra were calculated using the Datama software package (version 2.0) [72].

## 4. Conclusions

This comprehensive study of the molecular properties of water and gelatin in physically and chemically cross-linked gelatin hydrogels was fulfilled using FTIR-spectroscopy, NMR relaxation and diffusivity, and broadband dielectric spectroscopy, which are sensitive to the dynamical structure of water, a structural marker of polymer network structure. The applied experimental approaches demonstrated definite reinforcement of the hydrogel net structure and an increase in the quantity of hydrate water. FTIR experiments have shown that the chemical cross-linking of gelatin molecules results in an increase in the collagen-like triple helices’ “strength” as a result of the infused restriction of protein molecular mobility. The “strengthening” of protein chains hinders the mobility of protein fragments, thus introducing complex modifications to the water structural properties and retaining them from the disturbing action of temperature increases up to 30–40 °C.

## Figures and Tables

**Figure 1 ijms-25-11738-f001:**
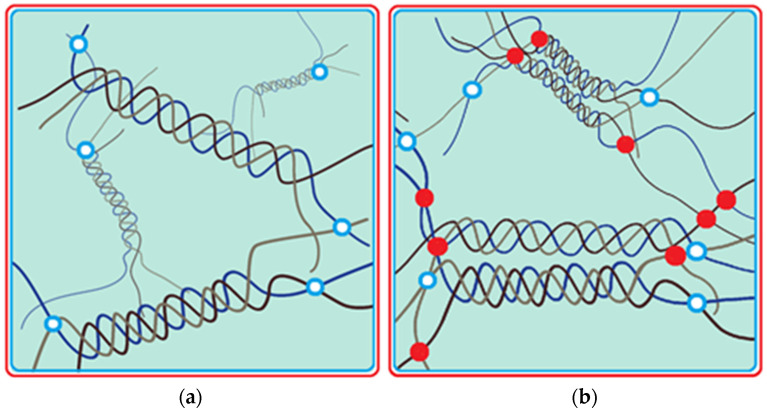
Schemes of gelatin network in “physical” (**a**) and ”chemical” (**b**) gels. Red circles show covalent bonding of gelatin chains, blue ones display hydrogen bonding.

**Figure 2 ijms-25-11738-f002:**
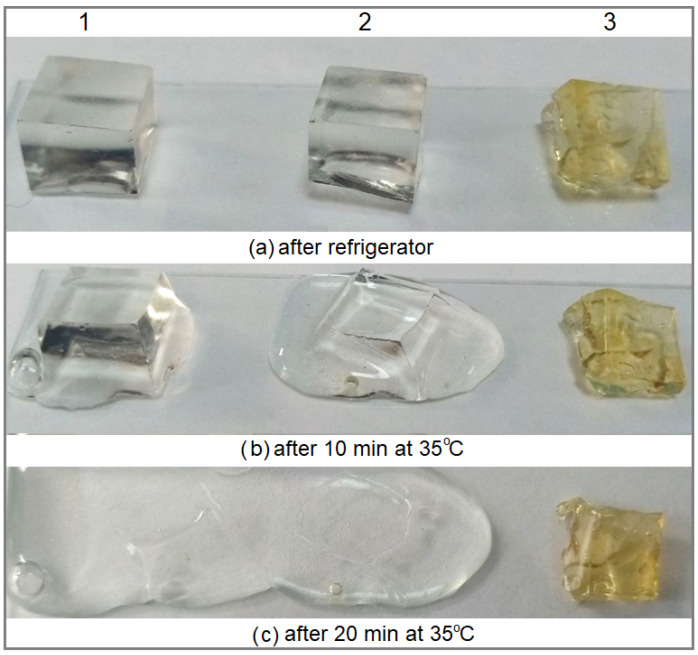
Thermal stability of gelatin hydrogels: original (1), original in the presence of 1% NaCl (2), and chemically cross-linked (3).

**Figure 3 ijms-25-11738-f003:**
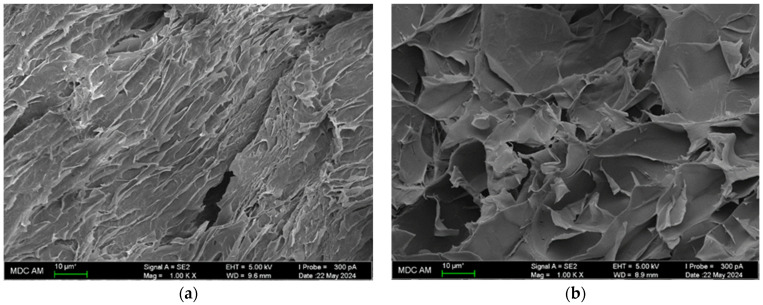
SEM images of “physical” (**a**) and “chemical” (**b**) gelatin hydrogels.

**Figure 4 ijms-25-11738-f004:**
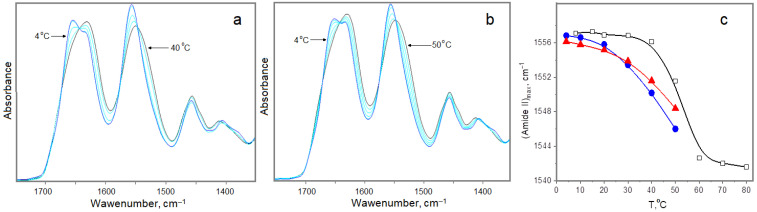
FTIR spectra in the absorption region of Amide I (C=O) and Amide II (N-H) for original (**a**) and chemically cross-linked (**b**) gelatin hydrogels, frequency position of Amide II maximum in spectra (**c**) of original (blue) and cross-linked (red) hydrogels and collagen from rat tail (black).

**Figure 5 ijms-25-11738-f005:**
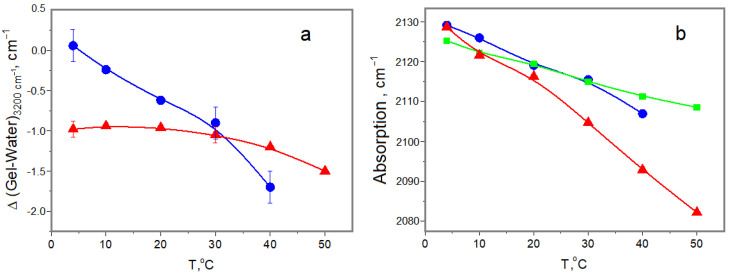
Temperature alterations in water stretching vibration at 3200 cm^−1^ (**a**) and water libration at 2130 cm^−1^ (**b**). Original (blue circles and cross-linked (red triangles) gelatin hydrogels and pure water (green).

**Figure 6 ijms-25-11738-f006:**
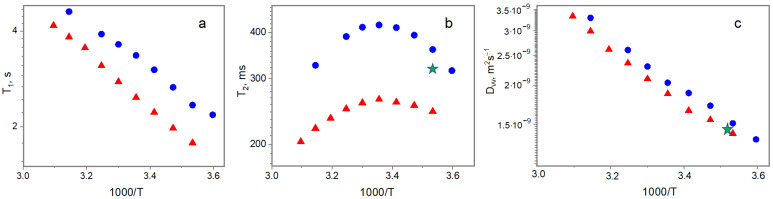
Temperature dependences of T_1_ (**a**), T_2_ (**b**), and D (**c**), characterizing mobility of water in original (blue circles) and cross-linked (red triangles) hydrogels and in original gel with water replaced by 1% NaCl water solution (green star).

**Figure 7 ijms-25-11738-f007:**
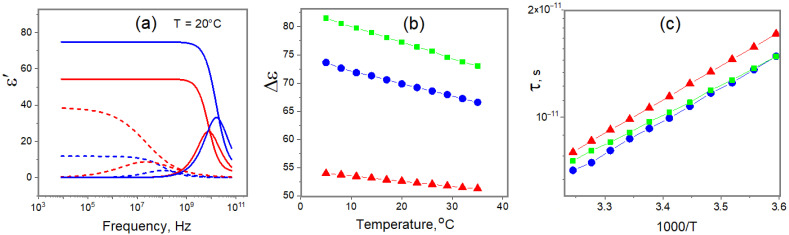
Fitting (dot lines) of primary experimental result (solid lines) by Equation (5) (**a**), temperature dependences of Δε (**b**) and τ (**c**) for water dielectric relaxation in the pure bulk (green), original (blue), and cross-linked (red) hydrogels.

**Figure 8 ijms-25-11738-f008:**
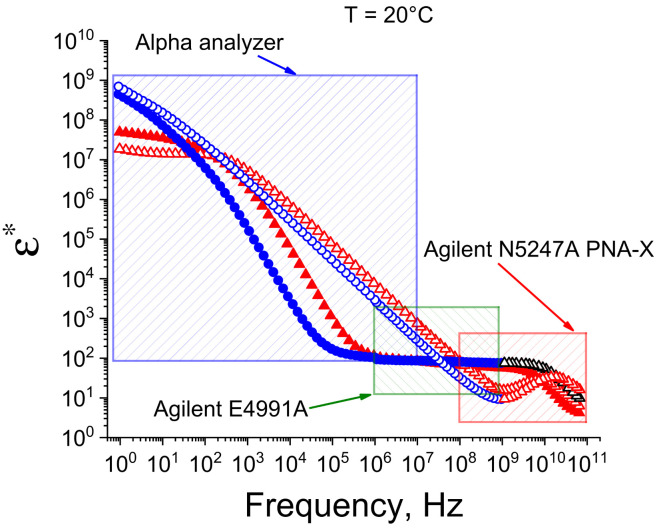
Experimental frequency ranges: red area—from 100 MHz to 60 GHz; green area—from 1 MHz to 1 GHz; blue area—from 1 Hz to 10 MHz. Original (blue) and cross-linked (red) hydrogels.

## Data Availability

All the data used for the analyses in this report are available from the corresponding author upon reasonable request.

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
