# Peer review of "Water as a Structural Marker in Gelatin Hydrogels with Different Cross-Linking Nature"

_ijms, 2024, doi:10.3390/ijms252111738_

Round 1
Reviewer 1 Report
Comments and Suggestions for Authors
They studied molecular properties of water and gelatin in physically and chemically cross-linked gelatin hydrogels by the FTIR-spectroscopy, NMR relaxation and diffusivity, broadband dielectric spectroscopy and conductivity techniques. The target matters are very popular ones. Therefore, very systematic research to discuss differences between physically and chemically cross-linked gelatin hydrogels is necessary to conclude meaningful insights and results. This work uses various spectral methods. However, the used samples are limited. They do not represent general features of physically and chemically cross-linked gelatin hydrogels. The presented data are very local. IR spectra do not clearly confirm evidences of chemical cross-linking. Cross-linking degrees are nor revealed, although it is crucial factors. Entirely, the presented data and the tested samples are very insufficient. Based on these features, conclusion is not so reliable. I do not recommend publication of this work in Int. J. Mol. Sci.
Author Response
Dear Reviewer, first of all our thanks to your time spent to our work and very helpful comments.
We do not agree with you that our target matter belongs to very popular ones. This new gelatin-based system was proposed a couple years ago and the are no corresponding studies which will help to explain molecular origin of novel properties of propose system. The question concerns of so-called “jelly ice cubes”, when gelatin was proposed for engineering of novel cooling systems, moreover carrying high antimicrobial activity. This technology involves a rapid-freezing slow-thawing treatment and a subsequent photo-crosslinking reaction. The developed freezing-thawing treatment results in formation of more uniform gelatin network before its chemical bonding. The use of preliminary physical cross-linking in combination with chemical bonding imparts stable mechanical properties to hydrogels, which ensure structural stability during temperature fluctuations and phase transitions of water. We supposed the corresponding alteration of gelatin and water properties under the changes in gelatin hydrogel structure and tried to find information on the difference in its properties in comparison with traditional gelatin hydrogel.
Dear Reviewer, we tried to give additional explanations of the novelty of our research in Introduction. And we supposed for the first results it enough to have only to systems – the novel one and control. We suppose that our experimental results show sufficiently clear the structural difference between two system. We agree with you, that IR spectra do not clearly confirm evidences of chemical cross-linking. It is not the subject of this method. From the other side FTIR study have clearly shown the alterations in protein structure and accompanied water properties as a result of chemical cross-linking.

Reviewer 2 Report
Comments and Suggestions for Authors
In this paper, the authors reported the molecular properties of gelatin hydrogels and then proposed that water is a marker of gel structural peculiarities.
This paper provided useful information and a novel concept. The experimental procedures were detailed, and the experimental data were presented clearly
However, this paper lacks quantitative analysis. Some conclusions were derived by visual methods, not quantitative data analysis.
1. “ Figure 6 shows temperature dependences of T1, T2 and D for both studied systems. For water protons with sufficient mobility one can consider the following correlation: T1, T2 ~ 1/τc, (1)”
Please use the data analysis to validate this assumption.
2. “ According to the previously obtained information on water properties in physical biopolymer hydrogels [35], T1 displays the water pool relaxation rates, which in the frame-work of two-site model [52,53] can be expressed as the weighted–average sum of contributions from mobility of free and bound water:
1/T1 = p/T1W-B + (1-p)/T1W-FREE, (2)”
Please derive the equation with theory or describe it clearly.
3. About the equation (3)
1/T2 = p1/T2W-B + p2/T2W-FREE + p3/T2-CROSS, (3)
Please derive the equation with theory or describe it clearly.
4. About the equation (5),
Please use the regression analysis to validate this equation.
5. The title could be revised. The terms “physical” and “chemical” could be deleted.
Comments on the Quality of English LanguageThe English needs to be improved to express the research.
Author Response
We thank dear Reviewer for his interest to our work and his time. Below we present our answers on comments:
- “ Figure 6 shows temperature dependences of T1, T2 and D for both studied systems. For water protons with sufficient mobility one can consider the following correlation: T1, T2 ~ 1/τc, (1)”
Please use the data analysis to validate this assumption.
Answer: We have revised this part of our manuscript, added specializing definitions and details and gave definite data analysis.
- “ According to the previously obtained information on water properties in physical biopolymer hydrogels [35], T1 displays the water pool relaxation rates, which in the frame-work of two-site model [52,53] can be expressed as the weighted–average sum of contributions from mobility of free and bound water:
1/T1 = p/T1W-B + (1-p)/T1W-FREE, (2)”
Please derive the equation with theory or describe it clearly.
Answer: we have revised the details of our explanation. First of all, let us comment that Eq.2, also as Eqs. 3 and for are follow from experimental features of NMR spectroscopy. The time-scale of experiment is much longer than the rates of molecular processes. Thus, in NMR many of measured parameters are the weighted–average sum of contributions from characteristics in different states.
- About the equation (3)
1/T2 = p1/T2W-B + p2/T2W-FREE + p3/T2-CROSS, (3)
Please derive the equation with theory or describe it clearly.
Answer: We have added necessary details. The explanation is just the same as in previous answer.
- About the equation (5),
Please use the regression analysis to validate this equation.
Answer: we have added the information that we used the least-squares procedure, applied to regression of experimental data to Eq.(5).
- The title could be revised. The terms “physical” and “chemical” could be deleted.
Answer: We revised the title of our manuscript. shows the result of least-squares procedure, applied to regression of experimental data.

Reviewer 3 Report
Comments and Suggestions for Authors
The paper "Molecular properties of “physical” and “chemical” gelatin hydrogels. Water as a marker of gel structural peculiarities" aims to highlight the role of water in strengthening the gel structure of gelatin. Both crosslinking pathways are shown, physically and chemically. The authors proved a very good documentation of State-of-the-art. Their results are supported by a combination of spectral methods: FT-IR, NMR, dielectric spectroscopy, conductivity.
I identified only minor issues as follows:
-Introduction section - the structure of gelatin must be presented along with the main interations intra- and intermolecular ones!
-what's new with this study?
- Section 2: Fig. 1 must be completed with the established interactions within the crosslinks;
- it is not clear how the authors have prepared the gels, physically and chemically!
-which is the crosslinking agent?
-Section 2.2.- IR interpretation is too general, the attribution of the main bands must be explained according to the physically or chemically modification of gelatin.
- H-bonds must be highlighted!
-a schematic reprezentation of the interactions with water molecules must be added!
-NMR spectra must be presented!
-what relevance has Fig. 8 for the experimental results?
-Conclusions- how the authors measure the quantity of hydrate water?
-
Author Response
We greatly appreciate the activity of dear Reviewer.
-Introduction section - the structure of gelatin must be presented along with the main interations intra- and intermolecular ones!
Answer: We have revised Introduction.
-what's new with this study?
Answer: We have studied not traditional cross-linked gelatin hydrogels. Information about the novel protocols and principally new gelatin system appears few years ago. Its authors proposed new system gelatin-base “jelly ice cubes”. There were found unique properties of novel system and proposed some suppositions for their molecular explanation. Our work is the first, where we try to find some explanations comparing novel gelatin hydrogel with the traditional one. So, we declare the absolute novelty of our research.
- Section 2: Fig. 1 must be completed with the established interactions within the crosslinks;
Answer: Sorry, we do not understand this comment. In the figure caption there is information: “Red circles show covalent bonding of gelatin chains, blue ones display hydrogen bonding”. There are only two main types of cross-links in the picture – covalent bonding and hydrogen bonding.
- it is not clear how the authors have prepared the gels, physically and chemically!
Answer: This information is presented in 3.1. Chemicals and sample preparations:
- To prepare original hydrogel aqueous solutions of 10% porcine gelatin was swelled at room temperature overnight, then was kept for 1 hour at 50 °C under ultrasonic treatment until being fully dissolved.
- To prepare chemical hydrogel, after that MSB solution was added to one of gelatin solution. The final solution, where the concentrations of MSB and gelatin were 1% and 10%, respectively, was mixed at 40 °C for 1 hour and used further to prepare “chemically” cross-linked gelatin hydrogel, according to known formulations
-which is the crosslinking agent?
Answer: photo-crosslinking reaction was induced by photosensitizer - menadione sodium bisulfite (MSB).
-Section 2.2.- IR interpretation is too general, the attribution of the main bands must be explained according to the physically or chemically modification of gelatin.
- H-bonds must be highlighted!
-a schematic reprezentation of the interactions with water molecules must be added!
Answers: thank you for these comments. We use FTIR techniques for long time for various water-biopolymer systems and suppose that there is enough corresponding information in the text.
-NMR spectra must be presented!
Answer: In this work we did not apply spectral analysis of NMR data. We mainly analyzed only dynamical information in from NMR experiments and suppose that it is not necessary to overload the manuscript with additional information.
-what relevance has Fig. 8 for the experimental results?
Answer: Fig.8 gives a common picture of complicated dielectric experiment. The dielectric relaxation processes usually are very broad on the frequency scale with large shifts from the system to system and with temperature. We gave Fig. 8 to demonstrate the necessary broad frequency range in dielectric experiment to register slow protein movements and fast relaxation of water in protein-water systems.
-Conclusions- how the authors measure the quantity of hydrate water?
Answer: There were no exact measurements of hydrate water. All experimental results, for example as decrease of relaxation times and self-diffusion coefficients was explained as overall increase of hydrate water. This is general practice and this result has many confirmations in published literature. At the same time, we made some simple estimations based on T1 data and presented some results on the quantity at lines 294-297.
Round 2
Reviewer 1 Report
Comments and Suggestions for Authors
It looks OK.